# DreamOn: Diffusion Language Models For Code Infilling Beyond Fixed-size Canvas

**Zirui Wu**[1,4*], **Lin Zheng**[1*], **Zhihui Xie**[1], **Jiacheng Ye**[1], **Jiahui Gao**[1], **Shansan Gong**[1],
**Yansong Feng**[4], **Zhenguo Li**[3], **Wei Bi**[2], **Guorui Zhou**[2], **Lingpeng Kong**[1†]
[1]The University of Hong Kong    [2]Kuaishou Technology    [3]Huawei Noah Ark Lab
[4]Peking University
ziruiwu@pku.edu.cn, lzheng2@cs.hku.hk, lpk@cs.hku.hk

## Abstract

Diffusion Language Models (DLMs) present a compelling alternative to autoregressive models, offering flexible, any-order infilling without specialized prompting design. However, their practical utility is blocked by a critical limitation: the requirement of a fixed-length masked sequence for generation. This constraint severely degrades code infilling performance when the predefined mask size mismatches the ideal completion length. To address this, we propose DreamOn, a novel diffusion framework that enables dynamic, variable-length generation. DreamOn augments the diffusion process with two length control states, allowing the model to autonomously expand or contract the output length based solely on its own predictions. We integrate this mechanism into existing DLMs with minimal modifications to the training objective and no architectural changes. Built upon Dream-Coder-7B and DiffuCoder-7B, DreamOn achieves infilling performance on par with state-of-the-art autoregressive models on HumanEval-Infilling and SantaCoder-FIM and matches oracle performance achieved with ground-truth length. Our work removes a fundamental barrier to the practical deployment of DLMs, significantly advancing their flexibility and applicability for variable-length generation. Our code is available at
https://github.com/DreamLM/DreamOn.

## 1 Introduction

In recent years, autoregressive language models have achieved remarkable progress (Comanici et al., 2025; OpenAI, 2025; Guo et al., 2025; Qwen et al., 2025). They model language as generating text sequentially in a fixed left-to-right manner. While dominant, this paradigm is now being complemented by Diffusion Language Models (DLMs) (Hoogeboom et al., 2021; Austin et al., 2021; Zheng et al., 2023; Singh et al., 2023; Lou et al., 2024; Sahoo et al., 2024; Shi et al., 2024; Nie et al., 2025; Ye et al., 2025; DeepMind, 2025; Labs et al., 2025), which have emerged as a promising alternative and are gaining significant attention.

DLMs operate through a multi-step denoising process, progressively refining a masked sequence to enable flexible, any-order generation (Austin et al., 2021; Hoogeboom et al., 2021). This property makes them inherently suited for infilling tasks—generating content to fill between a given prefix and suffix (Bavarian et al., 2022; Fried et al., 2023; Allal et al., 2023). In contrast, autoregressive models must resort to cumbersome workarounds for infilling, such as permuting the target span to the end of the sequence (Fried et al., 2023; Guo et al., 2024a; Hui et al., 2024; Seed et al., 2025). Such methods not only disrupt the natural contextual structure but also necessitate specialized prompting during training and infenence.

Despite the theoretical advantage, the practical application of DLMs is hindered by a critical bottleneck: the reliance on a pre-specified, fixed-length mask. Current DLMs (Ye et al., 2025; Nie et al., 2025; Xie et al., 2025; Gong et al., 2025b) require the input and output sequences to have

---

*Equal contribution.
†Corresponding author.

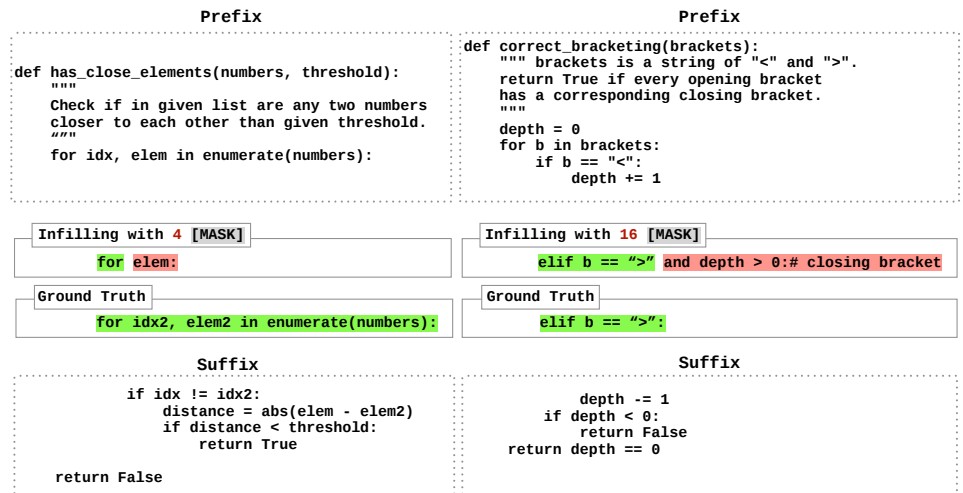

Figure 1: Example of DreamCoder-7B failing at code infilling due to the length mismatch between masked input and ground truth. Incorrect and correct code is marked in red and green. With too few masked tokens, diffusion models lack sufficient room for meaningful code infilling. Too many masks cause over-generation of unnecessary code snippet (e.g., `depth > 0` that is incorrect).

identical lengths, which prevents them from dynamically determining the length of the output. This limitation is especially damaging for code infilling, where solution lengths can vary significantly across examples. As shown in Figure 1, Dream-Coder-7B (Xie et al., 2025) produces incomplete or over-generated code when the mask length does not align with the ground truth. More critically, we observe an average performance drop of 38% on HumanEval-Infilling (Bavarian et al., 2022) when the predefined mask length does not align with the ground truth length (Table 2), highlighting the extreme sensitivity of current DLMs to this hyperparameter.

To address this bottleneck, we propose DREAMON, a discrete diffusion language modeling framework equipped with adaptive length adjustment (§3.1). DREAMON introduces dynamic length adaptation through two dedicated special tokens, [expand] and [delete], requiring no architectural modifications. We augment the standard diffusion training process with auxiliary length-control states, allowing DREAMON to be trained with minimal deviation from conventional DLM objectives (§3.2). During inference, the model adaptively expands and contract the masked sequence solely on its predictions without external guidance (§3.3). Based on Dream-Coder-7B (Xie et al., 2025) and DiffuCoder-7B (Gong et al., 2025b), DREAMON achieves competitive infilling performance with state-of-the-art autoregressive models on HumanEval-Infilling (Bavarian et al., 2022) and SantaCoder-FIM (Allal et al., 2023) (§4), and approaches oracle-level performance achieved with ground truth length (§5.1).

- We alleviate the fixed-length bottleneck of diffusion language models (DLMs) by introducing DREAMON, which allows the model to dynamically expand or contract sequences during generation without any architectural changes.
- Our method achieves variable-length generation with two special states [expand] and [delete], and supports scalable end-to-end learning of length adaptation through simple augmentation techniques with minimal deviation from standard diffusion objectives.
- On multiple infilling benchmarks, DREAMON delivers an average absolute performance boost of **26.4%** over diffusion baselines, matches the performance achieved with oracle length, and brings diffusion models close to or on par with state-of-the-art autoregressive models.

## 2 PRELIMINARY

Let $\mathbf{x}_0 = [\mathbf{x}_0^1, \ldots, \mathbf{x}_0^N]$ be a sequence of $N$ discrete tokens sampled from the data distribution $q(\mathbf{x})$. Each token takes values from a vocabulary of size $V + 1$, consisting of $V$ regular symbols plus an additional absorbing state [mask]. We represent each token $\mathbf{x}_{0,n}$, as well as the absorbing

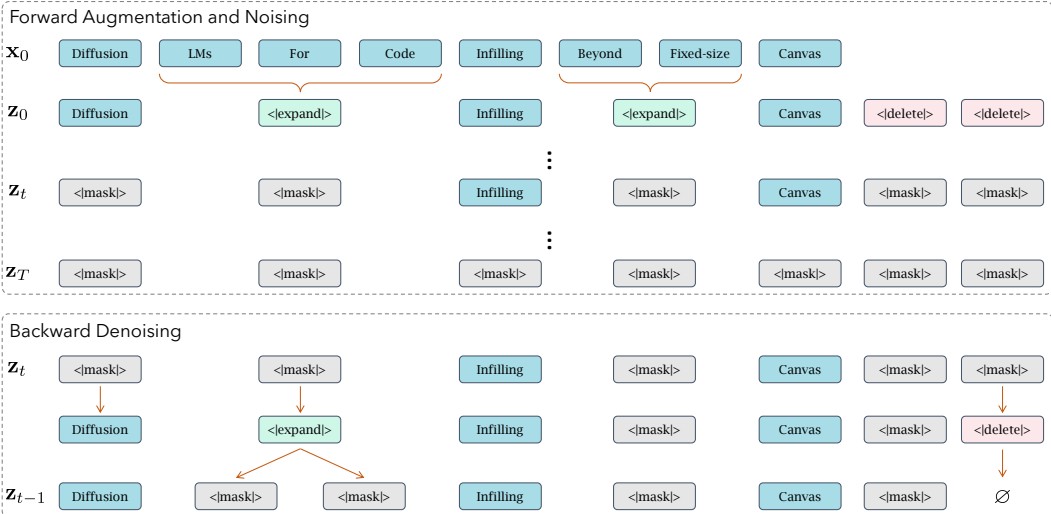

Figure 2: Overview of the augmented diffusion process. **Top**: the forward augmentation-and-noising procedure maps the input sequence $\mathbf{x}_0$ to an augmented latent $\mathbf{z}_0$ containing [expand] and [delete] states, and then applies a standard masked diffusion process over $\mathbf{z}_0$ to obtain $\mathbf{z}_t$ and eventually $\mathbf{z}_T$. **Bottom**: a single denoising step where [mask] positions in $\mathbf{z}_t$ can be predicted as either regular tokens or special states; [expand] deterministically expands into two [mask] tokens, while [delete] will remove the corresponding position, yielding a new sequence $\mathbf{z}_{t-1}$ with a different length from $\mathbf{z}_t$.

state [mask], as one-hot vectors in $\{0,1\}^{V+1}$. Typical discrete-time masked diffusion models are defined as a class of latent variable models over such sequences with a forward and backward transition process. In the forward process $q$, each token is preserved with a certain probability or replaced by [mask] otherwise, giving $q(\mathbf{x}_t \mid \mathbf{x}_0) = \alpha_t \mathbf{x}_0 + (1 - \alpha_t)$ [mask] with a predefined schedule $\alpha_t$. As $t$ increases, the schedule is designed such that the sequence converges to full mask tokens. The generative model reverses this process, starting from $\mathbf{x}_T$ and applying parameterized transitions $p_\theta(\mathbf{x}_{t-1} \mid \mathbf{x}_t)$ that approximate the true posterior $q(\mathbf{x}_{t-1} \mid \mathbf{x}_t, \mathbf{x}_0)$. This yields $p_\theta(\mathbf{x}_{0:T}) = p(\mathbf{x}_T) \prod_{t=1}^{T} p_\theta(\mathbf{x}_{t-1} \mid \mathbf{x}_t)$.

This class of generative models can be generalized to **continuous-time** parameterization by considering $t \in [0, 1]$, which avoids the bias introduced by predefined discretization over time steps. We adopt the frameworks in Kingma et al. (2021); Campbell et al. (2022); Sahoo et al. (2024); Shi et al. (2024); Ou et al. (2025) and train $p_\theta$ with a weighted cross-entropy loss objective,

$$\mathcal{L}(\theta) = -\mathbb{E}_{\substack{\mathbf{x}_0 \sim q(\mathbf{x}) \\ t \sim \mathcal{U}(0,1) \\ \mathbf{x}_t \sim q(\mathbf{x}_t | \mathbf{x}_0)}} \left[ w(t) \sum_{n=1}^{N} \mathbf{1}_{[\mathbf{x}_t^n = [\text{mask}]]} \log p_\theta(\mathbf{x}_0^n \mid \mathbf{x}_t) \right], \tag{1}$$

where the indicator $\mathbf{1}_{[\mathbf{x}_t^n = [\text{mask}]]}$ implies the loss is only evaluated on masked positions, and $w(t) \in (0, 1]$ is a time-dependent weighting term derived from the noise schedule $\alpha_t$ (Shi et al., 2024; Gong et al., 2025a). This objective provides a tractable variational upper bound on the negative log-likelihood and serves as an effective training target for large-scale diffusion language models.

## 3 METHOD

In this section, we present our formulation for extending masked diffusion models beyond fixed-length generation. We begin with an overview of our framework in §3.1, followed by training and inference procedures in sections 3.2 and 3.3, and practical implementation details in §3.4.

---

**Algorithm 1** DREAMON Training

---

**Require:** Model parameters $p_\theta$, merge rate scheduler $\mathcal{S}$;
 1: **repeat**
 2:     Sample original data $\mathbf{x}_0 \sim q(\mathbf{x})$ and a time step $t \sim \mathcal{U}(0, 1)$;
 3:     Construct augmented sequence $\mathbf{z}_0$ from $\mathbf{x}_0$ with [expand] and [delete] with $\mathcal{S}$;
 4:     Sample masked sequence $\mathbf{z}_t \sim q(\mathbf{z}_t|\mathbf{z}_0)$;
 5:     Compute weighted loss $\mathcal{L}(\theta)$ via eq. (2);
 6:     Update parameters $\boldsymbol{\theta}$ via $\nabla_{\boldsymbol{\theta}}\mathcal{L}$;
 7: **until** convergence

---

## 3.1 MASKED DIFFUSION WITH AUGMENTED STATES FOR LENGTH CONTROL

The key ingredient of DREAMON is to introduce two new special states [expand] and [delete] during the diffusion process. When a token transitions to the state [expand], we will expand it into two [mask] tokens at the same position of the sequence; and whenever a [delete] state is yielded, the token is removed from the sequence. With proper predictions of these special states, the model acquires native length control.

**Simulating Special Transitions via Data Augmentation.** To train the model to predict [expand] and [delete], we introduce an auxiliary augmented sequence $\mathbf{z}_0$ constructed from the original input $\mathbf{x}_0$. The augmentation merges random token spans into [expand] and inserts [delete] into the sequence. For merging, we first sample a time step $t \sim \text{Uniform}(0, 1)$ and compute a set of mask indices $\mathcal{M}_t$ according to the schedule $\alpha_t$. Rather than masking tokens, we use $\mathcal{M}_t$ to gate merging such that only spans of consecutive mask indices in $\mathcal{M}_t$ will be replaced with [expand], under a merging probability controlled by rate schedulers (§3.4). This pseudo-masking process provides finer control over the ratio of special to regular tokens, producing $\mathbf{z}_0$ with variable length and a balanced mix of regular and special states.

**Masked Diffusion over Augmented $\mathbf{z}_0$.** We impose the masked diffusion process $p_{\boldsymbol{\theta}}$ on $\mathbf{z}_0$. For these special states [expand] and [delete], the forward diffusion process always maps them to [mask], ensuring that all such tokens are masked and contribute to the learning signal. By construction, the prediction targets in the masked diffusion loss naturally include [expand] and [delete]. Consequently, the model is now trained to denoise not only regular tokens but also special sentinels, thereby learning length control behavior and enabling variable-length generation without any architecture changes.

## 3.2 TRAINING

Similarly to the masking state [mask] and any regular tokens, we treat [expand] and [delete] as sentinel tokens in the vocabulary, and the model is trained to predict them using the objective in Eq. (1). During training, however, we observe an imbalance: many [mask] positions correspond to [delete] targets, while far fewer correspond to [expand], since each [delete] is transformed into a single [mask], whereas multiple [mask] tokens are merged into one [expand]. As a result, [delete] tokens contribute disproportionately to the loss. To calibrate this, we introduce a loss weighting scheme that downscales the contribution of [delete] predictions so that their total weight is equivalent to that of a single [mask] prediction. The weighted training loss is then given by

$$\mathcal{L}(\theta) = -\mathbb{E}_{\substack{\mathbf{x}_0 \sim q(\mathbf{x}) \\ t \sim \mathcal{U}(0,1) \\ \mathbf{z_0} \sim q(\mathbf{z_0}|\mathbf{x_0}) \\ \mathbf{z} \sim q(\mathbf{z}_t|\mathbf{z}_0)}} \left[ w(t) \sum_{n=1}^{N} \mathbf{1}_{[\mathbf{z}_t^n = [\text{mask}]]} \cdot w_n \cdot \log p_\theta(\mathbf{z}_0^n \mid \mathbf{z}_t) \right], \tag{2}$$

with the per-token weight $w_n$ defined as

$$w_n = \frac{\mathcal{N}_{mask}}{\mathcal{N}_{mask} - \mathcal{N}_{delete} + 1} \times \begin{cases} 1, & \text{if } \mathbf{z}_0^n \neq [\text{delete}], \\ \dfrac{1}{\mathcal{N}_{delete}}, & \text{if } \mathbf{z}_0^n = [\text{delete}], \end{cases} \tag{3}$$

where $\mathcal{N}_{mask}$ and $\mathcal{N}_{delete}$ denote the number of [mask] and [delete] tokens in the sequence, respectively. The normalization factor ensures that the expected loss magnitude remains consistent across sequences with varying numbers of deletions.

## 3.3 INFERENCE

Our inference procedure, outlined in Algorithm 2, builds upon the standard masked diffusion denoising framework with key modifications to support variable-length generation. At each diffusion step, we simultaneously predict all masked positions and then selectively re-mask tokens based on prediction entropy, following Ye et al. (2025). However, the prior work employs fixed masking schedulers to determine how many tokens to unmask per step, which are ill-suited for dynamic-length modeling since they assume a pre-specified output length. Instead, we directly control the denoising trajectory by specifying $n$, the number of mask tokens to denoise at each step, enabling adaptive sequence length modeling. During denoising, predicted [expand] tokens are immediately expanded into two [mask] tokens, while generated [delete] tokens are removed from the sequence. To ensure stability and prevent unbounded growth, we enforce a maximum output sequence length $L_{max}$. The generation process terminates once all [mask] positions have been resolved.

---

**Algorithm 2** Variable-Length Generation with DREAMON

---

**Require:** Trained model parameters $\boldsymbol{\theta}$, initial sequence length $L$, maximum length $L_{\max}$, unmasking budget $n$ per iteration, and sampling temperature $\tau$;
1: **for** $l = 1, 2, \ldots, L$ **do**
2:     Initialize $\mathbf{z}^l \leftarrow$ [mask];
3: **end for**
4: **while** [mask] in $\mathbf{z}$ **do**
5:     Compute token probabilities $p \leftarrow p_{\boldsymbol{\theta}}(\cdot \mid \mathbf{z})$;
6:     **if** $|\mathbf{z}| \geq L_{\max}$ **then**
7:         Set the probability of [expand] to 0 and renormalize;
8:     **end if**
9:     Select up to $n$ masked positions with highest confidence;
10:     **for** each selected position $i$ **do**
11:         Draw $\widetilde{\mathbf{z}}^i \sim \text{Categorical}(p^i/\tau)$;
12:         **if** $\widetilde{\mathbf{z}}^i =$ [expand] **then**
13:             Replace $\mathbf{z}[i]$ with $[$[mask], [mask]$]$;
14:         **else if** $\widetilde{\mathbf{z}}^i =$ [delete] **then**
15:             Remove $\mathbf{z}[i]$ from the sequence;
16:         **else**
17:             Set $\mathbf{z}[i] \leftarrow \widetilde{\mathbf{z}}^i$;
18:         **end if**
19:         Update position indices if length has changed;
20:     **end for**
21: **end while**
22: **Return** $\mathbf{z}$.

---

## 3.4 PRACTICAL IMPLEMENTATIONS

**Span Merging Schedulers.** We design two empirical mask merging schedulers. (1) **Static scheduler**: merges adjacent [mask] tokens with a fixed probability $p_{merge}$ and (2) **Dynamic inverse scheduler**: sets the merging probability inversely proportional to the number of [mask] tokens in the sequence. This scheduler merges less with more [mask] tokens to avoid merging too many tokens that might potentially influence the original performance of the base model. We find that a mixture of two schedulers during training yield the best performance as detailed in §5.2.

**Broadcasting Deletion as Length Predictor.** In practice, we observe that performance degrades slightly when there is a large discrepancy between the initial masked span and the true target length. This introduces inefficiency during inference, as the model must expend numerous forward passes

to adjust the sequence length via incremental expansions or contractions. To mitigate this, we introduce a training-free inference-time adaptation method that accelerates convergence. Specifically, whenever the model predicts a `[delete]`, we eliminate all subsequent tokens to its right if they are all `[mask]` tokens. This mechanism significantly reduces unnecessary computation and improves inference efficiency without sacrificing generation quality.

## 4 EXPERIMENTS

### 4.1 SETUP

We fine-tune Dream-7B (Ye et al., 2025), DiffuCoder-7B (Gong et al., 2025b), and DreamCoder-7B (Xie et al., 2025) on the education-instruction subset of OpenCoder SFT data (Huang et al., 2024), which contains about 110K Python instruction-solution pairs synthesized from high-quality educational data. Our experiments focus on code infilling, where the goal is to generate missing spans conditioned on surrounding prefix and suffix contexts. During training, we randomly split each solution into three segments: prefix, middle, and suffix. The instruction, prefix, and suffix are fixed as context, while diffusion is applied only to the middle segment.

For sequence contraction, we find it sufficient to append a random number of `[delete]` tokens (from 0 to 64) to the end of the middle segment during training. For sequence expansion, `[expand]` tokens are constructed with merging probability $p_{\mathrm{merge}}$ as 0.5, using a 1:1 mix of static and dynamic inverse schedulers. Models are trained for 10 epochs with batch size 128, maximum context length 1024, and learning rate $1e-5$ under a cosine decay schedule with 10% warmup steps. It takes approximately 5 hours to train with 8 H800 GPUs. The compute of DreamOn is only 0.15% compared with the compute for pretraining a base model (Ye et al., 2025). During inference we set temperature as 0.2 and top_p as 0.9. To prevent excessive growth, we cap mask expansion in DREAMON at $L_{max} = 128$. We also disable mask expansion in inference after expanding $L_{max}$ times.

### 4.2 EVALUATION

**Baselines.** We compare against state-of-the-art *autoregressive* models pretrained with infilling objectives, specifically Deepseek-Coder-6.7B (Guo et al., 2024b), Qwen2.5-Coder-7B (Hui et al., 2024) and Seed-Coder-8B (Seed et al., 2025). For open-source *diffusion* language model baselines of similar scale, we evaluate LLaDA-8B (Nie et al., 2025), Dream-7B (Ye et al., 2025), DiffuCoder-7B (Gong et al., 2025b), and DreamCoder-7B (Xie et al., 2025).

**Benchmarks.** We evaluate models on HumanEval-Infilling (Bavarian et al., 2022) benchmarks, including single-line and multi-line subsets, and the Python subset of Santacoder-FIM (Allal et al., 2023). We use the official evaluation scripts to report pass@1 for HumanEval-Infilling and exact match for Santacoder-FIM. We evaluate autoregressive language models using their respective infilling templates used during pretraining. For all diffusion models, we set the mask length to 64 by default. DREAMON variants dynamically adjust this length as detailed in §3.4.

### 4.3 RESULTS

Table 1 shows that baseline diffusion models struggle with code infilling due to their fixed-length generation, lagging significantly behind autoregressive models. DREAMON effectively resolves this limitation, yielding an average absolute improvement of 26.4% over diffusion baselines and highlighting its effectiveness as a *model-agnostic* enhancement.

Notably, with DREAMON, DiffuCoder-7B and DreamCoder-7B not only match the performance of leading autoregressive models like Qwen2.5-Coder-7B, but also surpasses them in the more challenging multi-line infilling benchmark. This demonstrates that equipping diffusion models with our length-adaptive mechanism makes them highly competitive for infilling tasks.

Table 1: Pass@1 on HumanEval-Infilling and exact match on Santacoder-FIM, comparing open-source auto-regressive and diffusion model baselines.The best results across diffusion models are shown in bold, and the second best are underlined.

| Models | HumanEval-Infilling (Pass@1) | | SantaCoder (EM) |
| --- | --- | --- | --- |
| | Single-line | Multi-line | |
| **Open-Weights AR Models** | | | |
| Deepseek-Coder-6.7B | 73.0 | 45.7 | 76.3 |
| Seed-Coder-8B | 89.7 | 59.3 | 77.2 |
| Qwen2.5-Coder-7B | 92.6 | 58.7 | 79.8 |
| **Open-Weights Diffusion Models** | | | |
| LLaDA-8B | 48.3 | 21.1 | 35.1 |
| Dream-7B | 48.2 | 21.9 | 60.3 |
| + DREAMON | $88.6_{+40.4}$ | $53.3_{+31.4}$ | $73.8_{+13.5}$ |
| DiffuCoder-7B | 53.7 | 45.0 | 58.0 |
| + DREAMON | $\textbf{92.2}_{+38.5}$ | $63.1_{+18.1}$ | $\underline{77.4}_{+19.4}$ |
| DreamCoder-7B | 55.5 | 43.2 | 59.3 |
| + DREAMON | $\underline{92.1}_{+36.6}$ | $\textbf{63.8}_{+20.6}$ | $\textbf{79.0}_{+19.7}$ |

Table 2: Infilling performance across different designs for diffusion language models. *Oracle*: performance with the oracle target length for reference. †: We use an AST parser to compute exact match to normalize huge syntactic differences between the model output and the ground truth.

| Models | Initial Mask Length | | | | | Avg. | Oracle |
| --- | --- | --- | --- | --- | --- | --- | --- |
| | 4 | 8 | 16 | 32 | 64 | | |
| Single-Line (Pass@1) | | | | | | | |
| Dream-Coder-7B | 24.9 | 61.2 | 72.6 | 62.4 | 55.5 | 55.3 | 93.3 |
| + DREAMON | **88.7** | **90.6** | **91.0** | **91.6** | **92.1** | **90.8** | 91.6 |
| w/o Delete | 87.8 | 77.9 | 71.2 | 62.3 | 37.8 | 67.4 | 93.3 |
| w/o Expand | 25.1 | 71.6 | 88.0 | 90.9 | 91.5 | 73.4 | 92.5 |
| Multi-line (Pass@1) | | | | | | | |
| Dream-Coder-7B | 5.5 | 14.7 | 27.1 | 39.4 | 43.2 | 26.0 | 69.0 |
| + DREAMON | **50.2** | **53.8** | **56.9** | **60.9** | **63.8** | **57.1** | 66.6 |
| w/o Delete | 44.6 | 45.3 | 46.1 | 46.7 | 44.7 | 45.5 | 67.9 |
| w/o Expand | 5.5 | 16.5 | 30.7 | 48.2 | 61.3 | 32.4 | 63.2 |
| SantaCoder-FIM (EM) | | | | | | | |
| Dream-Coder-7B | 20.0 | 26.6 | 43.5 | 50.8 | 59.3 | 40.0 | 76.3† |
| + DREAMON | **75.0** | **76.8** | **78.4** | **78.0** | **79.0** | **77.4** | 82.0 |
| w/o Delete | 74.2 | 44.3 | 40.2 | 50.0 | 56.2 | 53.0 | 84.2 |
| w/o Expand | 22.5 | 55.0 | 74.7 | 77.8 | 78.0 | 61.6 | 78.6† |

## 5 ANALYSIS

In this section, we conduct ablation studies to evaluate the effectiveness of DREAMON. All variants are fine-tuned from DreamCoder-7B and evaluated with initial mask lengths ranging from 4 to 64. We additionally evaluate the infilling results under an *oracle* setting, where the initial mask length matches the ground-truth solution length, providing an approximate upper bound for infilling performance of diffusion language models.

## 5.1 PERFORMANCE WITH DIFFERENT MASK LENGTHS

**Performance Breakdown.** As shown in Table 2, DreamCoder-7B without finetuning suffers significant performance degradation when using fixed mask lengths compared to the oracle-length performance, highlighting the strong dependence of infilling quality on accurate mask length. By contrast, DREAMON achieves near oracle-level performance across a wide range of initial mask lengths. Importantly, the performance gains stem from the combined use of both mask expansion and contraction mechanisms. DREAMON maintains stable performance on both single-line infilling and SantaCoder-FIM tasks regardless of the initial mask length. We provide two denoising trajectory examples in Appendix D.

**Ablation on Length Control.** To isolate the contributions of expansion and deletion mechanisms, we evaluate ablated variants of DREAMON: (1) **w/o Expand**, disabling mask expansion; and (2) **w/o Delete**, disabling mask deletion. Removing deletion leads to a sharp performance drop on longer mask lengths, as the model tends to over-generate and fill all given `[mask]` tokens. On the other hand, removing expansion severely harms performance on short lengths, as the model cannot dynamically extend mask sequences to accommodate more complex or longer completions. We also observe a slight performance decline on long masks without expansion, suggesting that even for longer masked inputs, expansion remains beneficial by allowing fine-grained length adjustments.

## 5.2 EXPANSION MECHANISM DESIGN

Mask merging strongly affects the number of `[expand]` tokens and the gap between initial and target sequence lengths. We study this in-depth and evaluate two merge rate schedulers: a static scheduler with fixed merge probability and a dynamic inverse scheduler with merge probability inversely proportional to the number of `[mask]` tokens. Using only the static scheduler enables effective expansion, achieving an 88.9% pass rate for length-4 masks. However, its performance is limited on longer masks.

The dynamic inverse scheduler merges less when more masks are present. It achieves higher performance on longer masks but struggles with large expansions, dropping to 82.5% on length-4 masks. We find a 1:1 mixture achieves the best overall results, offering a favorable balance across various mask lengths (Appendix C).

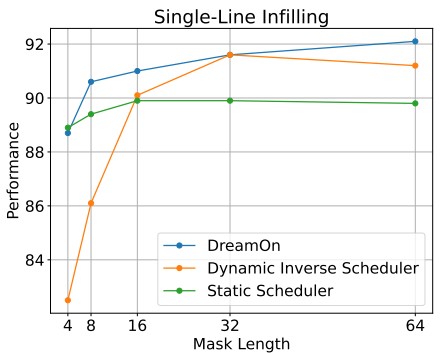

Figure 3: Ablation on merging rate scheduler design choices.

## 5.3 DELETION MECHANISM DESIGN

We ablate our design choices for handling `[delete]` tokens with the following experiments: (1) **w/o Loss Balancing**: train the model without down-weighing the loss on `[delete]` tokens, treating them equally with other tokens in the loss computation; (2) **w/o In-place Deletion**: Instead of removing `[delete]` tokens, keep them in the sequence, similar to generating padding placeholder tokens in standard diffusion language (Nie et al., 2025; Ye et al., 2025). To implement this, we randomly mask or preserve `[delete]` during training; and (3) **w/o Deletion Broadcasting**: disable the inference-time mechanism described in §3.4.

As shown in Table 2, removing loss balancing leads to a substantial performance drop to 84.6% average pass@1 rate, confirming that down-weighing `[delete]` loss is essential to prevent the model from overfitting to deletion signals. Keeping persistent `[delete]` tokens also performs poorly (average 85.3%), indicating that placeholder-like deletion tokens in the sequence disrupt positional coherence and degrade training. Disabling deletion broadcasting reduces performance by 0.6% on average, especially when the given mask length is much longer than the expected solution. The deletion broadcasting mechanism also accelerates generation by 2.1×.

Table 3: Ablation study for mask deletion mechanism implementations.

| Models | Initial Mask Length | | | | | Avg. | Oracle |
|---|---|---|---|---|---|---|---|
| | 4 | 8 | 16 | 32 | 64 | | |
| HumanEval-Infilling Single-Line (Pass@1) | | | | | | | |
| DREAMON | **88.7** | **90.6** | **91.0** | **91.6** | **92.1** | **90.8** | 91.6 |
| w/o Loss Balancing | 75.8 | 82.5 | 87.0 | 87.2 | 90.4 | 84.6 | 88.6 |
| w/o In-Place Deletion | 85.9 | 85.7 | 88.5 | 84.8 | 78.0 | 84.6 | 93.1 |
| w/o Deletion Broadcasting | **88.7** | 90.5 | 90.0 | 90.2 | 91.4 | 90.2 | 91.6 |

## 5.4 EFFICIENCY ANALYSIS OF DELETION BROADCASTING

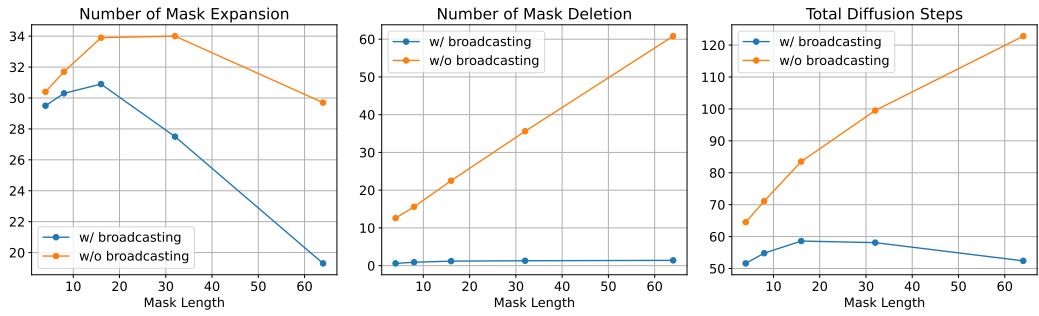

Figure 4: Average generation steps of DreamCoder + DREAMON on multi-line infilling subset.

The introduction of broadcasting dramatically enhances inference efficiency, primarily by transforming the deletion process from a token-by-token operation into a length prediction action. Without broadcasting, deletion steps scale almost linearly with the initial mask length because the model must iteratively predict and remove each excess mask token individually. In contrast, DREAMON with deletion broadcasting mechanism keep the number of mask deletion nearly constant around with roughly only 1 step on average. This optimization eliminates the computational bottleneck caused by large discrepancies between the initial masked span and the true target length, reducing total inference steps from as high as 122.8 (w/o broadcasting) to just 52.4 (w/ broadcasting) at mask length 64. Consequently, broadcasting not only slashes unnecessary forward passes but also stabilizes overall inference cost, making the generation process both faster and more robust to initialization variance without any impact on output quality.

## 6 RELATED WORK

**Code Infilling with Autoregressive Models.** Code infilling requires generating missing code segments conditioned on bidirectional context, a task inherently challenging for standard left-to-right autoregressive models. To address this, several approaches adapt architectures to better capture bidirectional dependencies Yang et al. (2019); Stern et al. (2019); Gu et al. (2019a); Chan et al. (2019); Welleck et al. (2019); Shen et al. (2020); Alon et al. (2020); Nguyen et al. (2023); Shen et al. (2023).

A widely adopted alternative preserves the standard left-to-right autoregressive architecture by relocating the target infill segment to the end of the input sequence, enabling the model to generate the missing code autoregressively (Raffel et al., 2020; Tay et al., 2023; 2022; Anil et al., 2023; Sun et al., 2024). This approach is compatible with decoder-only architectures (Bavarian et al., 2022) and has become the standard in modern code language models, including Codex (OpenAI et al., 2022), INCODER (Fried et al., 2023), SANTACODER (Allal et al., 2023), StarCoder (Li et al., 2023) and StarCoder 2 (Lozhkov et al., 2024), CODEGEN 2/2.5 (Nijkamp et al., 2023), Code-Llama (Roziere et al., 2023), DeepSeek-Coder (Guo et al., 2024a), CodeGemma (CodeGemma Team, 2024), Qwen-Coder (Bai et al., 2023; Hui et al., 2024), and Seed-Coder (Seed et al., 2025).

**Discrete Diffusion Language Models.** Discrete diffusion models have recently emerged as a compelling alternative to autoregressive models. Foundational work by Austin et al. (2021); Hoogeboom et al. (2021) introduced discrete diffusion processes for text data, enabling probabilistic modeling of token sequences through iterative and bidirectional denoising. Subsequent research has refined these approaches with continuous-time relaxations (Campbell et al., 2022), improved training objectives (Zheng et al., 2023; Lou et al., 2024), and generalized masked diffusion frameworks (Sahoo et al., 2024; Shi et al., 2024; Ou et al., 2025). Scaling efforts have produced powerful models such as Plaid (Gulrajani & Hashimoto, 2023) and LLaDA (Nie et al., 2025). Adaptation techniques leveraging pretrained models, such as DiffuLLaMA (Gong et al., 2025a) and Dream (Ye et al., 2025), have narrowed the performance gap with state-of-the-art autoregressive language models.

**Non-autoregressive Models with Length Control.** Generating variable-length sequences remains a significant challenge for non-autoregressive models. Prior work has explored diverse strategies to address this, including learning separate length predictors (Gu et al., 2018; Lee et al., 2018; Ghazvininejad et al., 2019; Zheng et al., 2023), marginalizing over latent alignments to contract sequence length (Chan et al., 2020), incorporating edit operations (Gu et al., 2019a;b; Stern et al., 2019; Johnson et al., 2021; Reid et al., 2023; Campbell et al., 2023; Patel et al., 2025; Havasi et al., 2025), and performing diffusion over sequence positions (Zhang et al., 2025; Kim et al., 2025).

Recent concurrent works also address this challenge. Edit flows (Havasi et al., 2025) present a discrete flow matching with edit operations over extended spaces for tractable and effective training; DDOT (Zhang et al., 2025) proposes to jointly denoise token states and positions for dynamic segment-length adjustment; FlexMDM (Kim et al., 2025) learns insertion and unmasking rates through a joint interpolant framework over both token states and positions, thereby enabling variable-length generation; and DAEDAL (Li et al., 2025) provides a training-free approach using inference-time prediction confidence scores to dynamically determine the response length.

In contrast, our method implements native length control in masked diffusion models with minimal additional training and no architectural modifications, directly adapting pretrained diffusion language models. This design preserves the simplicity of the original model, avoids complicated multi-stage inference pipelines, and yields substantial gains in flexibility and performance for variable-length generation.

## 7 CONCLUSIONS

In this work, we introduce DREAMON, a simple yet effective framework that enables dynamic length control through two special tokens (`[delete]` and `[expand]`) without architectural changes. By augmenting the diffusion process with auxiliary length-control states, DREAMON learns to expand or contract sequences based solely on model confidence. Our results show that DREAMON approaches oracle-length performance and achieves competitive results with state-of-the-art autoregressive models. We hope our work can pave the way for more practical and flexible DLMs beyond fixed-size canvas.

**Limitations.** Currently, we limit our evaluation to focus on code infilling tasks that require strong variable-length generation capabilities. Future work will extend the scope to broader applications to assess the generalizability of DREAMON. In addition, the training and inference procedures in DREAMON rely on heuristics to enable variable-length generation in a simple yet effective manner; developing a more principled formulation for flexible inference in masked diffusion models is an important direction for future research. Finally, our current design uses a single expansion state `[expand]` that deterministically expands into two `[mask]` tokens. This choice keeps the output space small and training stable, but requires multiple expansion steps when the target completion is much longer than the initial mask span. A promising direction for future work is to introduce a richer vocabulary with multiple expansion factors or to couple expansion with an explicit length-prediction head. They could reduce the number of denoising iterations, but would also enlarge the decision space and require careful rebalancing of the training objective to maintain well-behaved length adjustment dynamics. We leave the exploration of these richer expansion schemes to future work.

## ACKNOWLEDGEMENTS

We acknowledge the open-source community for providing high-quality datasets and evaluation frameworks. This research was supported in part by the joint research scheme of the National Natural Science Foundation of China (NSFC) and the Research Grants Council (RGC) under grant number N_HKU714/21.

## ETHICS STATEMENT

Our research focuses on developing a diffusion-based language modeling method capable of variable-length text generation. We did not collect any data involving human subjects, private information. And our study does not include any human evaluation. All datasets used in our experiments are publicly available benchmarks, and we strictly adhere to their respective usage licenses. Furthermore, our method does not present any foreseeable risks of misuse or societal harm.

## REPRODUCIBILITY STATEMENT

We have taken deliberate steps to ensure the reproducibility of our work. Detailed descriptions of the experimental setups and hyperparameter configurations are provided in §4.

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

## A  THE USE OF LARGE LANGUAGE MODELS

We employ large language models primarily for polishing written text—for example, to correct grammar and improve clarity. However, LLMs do not play a significant role in the core research activities, including idea generation, experimental design, or the substantive writing of the manuscript.

## B  GENERALIZABILITY BEYOND CODE INFILLING

To demonstrate that DREAMON is not limited to code infilling, we further evaluate the model's capability of commonsense narrative understanding on the ROCStories corpus Mostafazadeh et al. (2016). We fine-tune Dream-7B (Ye et al., 2025) on the ROCStories training split in two variants: with DREAMON, and an SFT baseline trained following the recipe in Ye et al. (2025). We consider two setups: (1) **Narrative Infilling**, where the model generates a missing sentence within the middle of a story, requiring bidirectional context understanding; and (2) **Prefix-guided Generation**, where the model is provided with the preceding story context and generates the final concluding sentence, serving as a proxy for general completion tasks.

Table 4 presents Rouge-L scores across varying initial mask lengths. In both infilling and prefix-guided settings, the baseline performance degrades significantly when there is a mismatch between the initialized mask length and the natural length of the missing content (e.g., at lengths 4 and 32). In contrast, DREAMON utilizes its length-adaptive logic—mediated by `[expand]` and `[delete]`-to achieve superior performance.

Crucially, our method exhibits high stability: the generation quality remains consistent regardless of the initial mask length. This confirms that the proposed length-adaptation mechanism successfully decouples generation quality from the initial mask length, demonstrating robust generalizability to creative, variable-length natural language tasks.

Table 4: Rouge-L scores on the ROCStories corpus across variable initial mask lengths.

| Method | Initial Mask Length | | | |
|---|---|---|---|---|
| | 4 | 8 | 16 | 32 |
| Narrative Infilling | | | | |
| Dream + SFT | 19.2 | 29.8 | 26.5 | 18.9 |
| **DREAMON** | **31.6** | **31.4** | **31.3** | **30.6** |
| Story Ending Generation | | | | |
| Dream + SFT | 16.3 | **25.1** | 22.4 | 16.7 |
| **DREAMON** | **24.5** | 24.6 | **24.4** | **24.1** |

## C   ABLATION FOR HYPERPARAMETERS

In this section,we provide the results on DreamCoder-7B with different training hyperparameters. As shown in Figure 5a, the Pass@1 score for single-line infilling reaches its peak—approximately 90.9%—when employing a balanced 1:1 mixture of static and dynamic inverse schedulers. This result highlights the substantial performance gain achieved through this synergistic combination. Similarly, the right panel reveals that the model attains its highest Pass@1 score of roughly 90.5% at a merge probability of 0.5. Guided by these findings, we adopt a 1:1 static/dynamic scheduler mix ratio and a merge probability of 0.5 in DREAMON configuration to maximize performance.

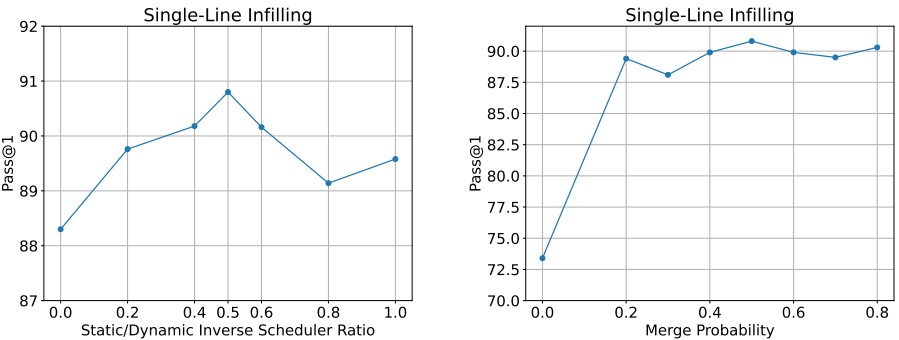

(a) Result with different scheduler merging ratio. (b) Result with different merging probability $p_{merge}$

Figure 5: Performance on single-line subset of HumanEvalInfilling-FIM with different hyperparameters during training. The performance is computed as the average pass@1 with mask length 4, 8, 16, 32 and 64.

## D   INFILLING EXAMPLES

A key advantage of DREAMON lies in its adaptive handling of sequence length variations during inference. This is achieved through two complementary states [expand] and [delete]. First, as depicted in Figure 6, DREAMON possesses the capability to expand mask sequences. This dynamic expansion allows the model to generate outputs longer than its initial input mask, effectively preventing truncation and enabling the generation of comprehensive sequences. Second, Figure 7 showcases the efficacy of the deletion broadcasting mechanism. This mechanism plays a crucial role

```
def has_close_elements(numbers, threshold):
    """
    Check if in given list are any two numbers
    closer to each other than given threshold.
    """
    for idx, elem in enumerate(numbers):
```
**Prefix**

```
        <|mask|><|mask|><|mask|><|mask|>
        <|expand|><|mask|><|mask|><|mask|>
        <|mask|><|mask|><|mask|><|mask|><|mask|>
        <|mask|><|expand|><|mask|><|mask|><|mask|>
        <|mask|><|mask|><|mask|><|mask|><|mask|><|mask|>
        <|mask|><|expand|><|mask|><|mask|><|mask|><|mask|>
        <|mask|><|mask|><|mask|><|mask|><|mask|><|mask|><|mask|>
        <|mask|><|mask|><|expand|><|mask|><|mask|><|mask|><|mask|>
        <|mask|><|mask|><|mask|><|mask|><|mask|><|mask|><|mask|><|mask|>
            <|mask|><|mask|><|mask|><|mask|><|mask|><|mask|><|mask|>
            <|mask|><|mask|><|expand|><|mask|><|mask|><|mask|><|mask|>
            <|mask|><|mask|><|mask|><|mask|><|mask|><|mask|><|mask|><|mask|>
            for<|mask|><|mask|><|mask|><|mask|><|mask|><|mask|><|mask|>
            for<|mask|>2<|mask|><|mask|><|mask|><|mask|><|mask|>
            for idx2<|mask|><|mask|><|mask|><|mask|><|mask|>
            for idx2<|mask|><|mask|><|mask|><|mask|>
            for idx2 elem<|mask|><|mask|><|mask|>
            for idx2 elem<|mask|><|expand|><|mask|>
            for idx2 elem<|mask|><|mask|><|mask|><|mask|>
            for idx2 elem2<|mask|><|mask|><|mask|>
            for idx2 elem2 in<|mask|><|mask|>
            for idx2 elem2 in<|mask|><|expand|>
            for idx2 elem2 in<|mask|><|mask|><|mask|>
            for idx2 elem2 in enumerate<|mask|><|mask|>
            for idx2 elem2 in enumerate(numbers<|mask|>
            for idx2 elem2 in enumerate(numbers):
```
**Infilling**

```
            if idx != idx2:
                distance = abs(elem - elem2)
                if distance < threshold:
                    return True

    return False
```
**Suffix**

Figure 6: DREAMON adds mask tokens as needed.

in promoting rapid convergence to the optimal predicted sequence length by selectively removing redundant mask tokens, thereby streamlining the generation process and improving efficiency.

```
def correct_bracketing(brackets: str):
    """ brackets is a string of "<" and ">".
    return True if every opening bracket has a corresponding closing bracket.
    """
    depth = 0
    for b in brackets:
        if b == "<":
            depth += 1
```
**Prefix**

```
<|mask|><|mask|><|mask|><|mask|><|mask|><|mask|><|mask|><|mask|>
<|mask|><|mask|><|mask|><|mask|><|mask|><|mask|><|mask|><|mask|>

<|mask|><|mask|><|mask|><|mask|><|mask|> ">|mask|><|mask|><|mask|>
<|mask|><|mask|><|mask|><|mask|><|mask|><|mask|><|mask|>

<|mask|><|mask|><|mask|><|mask|> "><|mask|><|mask|><|mask|><|mask|>
<|mask|><|mask|><|mask|><|mask|><|mask|><|mask>

<|mask|>elif<|mask|><|mask|> "><|mask|><|mask|><|mask|><|mask|>
<|mask|><|mask|><|mask|><|mask|><|mask|><|mask|>

<|mask|>elif<|mask|><|mask|> "><|mask|><|delete|><|mask|>
<|mask|><|mask|><|mask|><|mask|><|mask|><|mask|><|mask|>

        elif<|mask|><|mask|> "><|mask|>

        elif<|mask|> == ">"<|mask|>

        elif<|mask|> == ">":

        elif b == ">":
```
**Infilling**

```
            if idx != idx2:
                distance = abs(elem - elem2)
                if distance < threshold:
                    return True

    return False
```
**Suffix**

Figure 7: DREAMON deletes excess mask tokens with the deletion broadcasting mechanism.

