# OpenReview forum: "DreamOn: Diffusion Language Models For Code Infilling Beyond Fixed-size Canvas"
_ICLR.cc/2026/Conference — ICLR 2026 Poster_

### Official Review · Reviewer_SUPh · 2025-10-30

**Soundness:** 3
**Presentation:** 3
**Contribution:** 2
**Rating:** 4
**Confidence:** 3

**Summary:**

This paper solves the fixed-length mask limitation of code infilling diffusion models by introducing [expand] and [delete] tokens. The training process is adapted to first (synthetically) introduce these tokens in the sequence before embedding and predicting each token. Experiments show that diffusion models with these tokens perform significantly better than the original models, and even perform almost on par with auto-regressive models with infilling objective (pass @ 1).

**Strengths:**

- Controlling the size of the generation is a fundamental challenge in discrete diffusion models for text. Approaches that do not require (large) architectural changes can leverage pre-trained models more effectively.
- The ablations focus on relevant design decisions, such as not using either of the tokens.
- The paper is properly structured, well-written and clear.

**Weaknesses:**

- The paper mentions but does not compare against other approaches. FlexMDM and DDOT are both applicable to infilling and can be directly compared with, and their code is available.
- The paper considers only pass@1. CodeFusion [1—missing reference] shows that code diffusion models excel at higher pass@k rates, because of the model's increased diversity: compiler and execution feedback can be used to select candidates.

[1] Singh, M., Cambronero, J., Gulwani, S., Le, V., Negreanu, C. S., & Verbruggen, G. (2023, November). Codefusion: A pre-trained diffusion model for code generation. In The 2023 Conference on Empirical Methods in Natural Language Processing.

**Questions:**

- How does the performance evolve in function of the length of the final generation? I'm supposing that longer masks, which require more [insert], will perform worse?
- If the above holds, would extending [insert] into [insert1], [insert2], ..., [insertX] make the process converge in fewer iterations? In table 2, the multi-line performance has a big jump to the oracle performance, indicating that requiring many insertions is a bottleneck for longer code snippets.

---

> ### Author Response · Authors · 2025-11-21
> **Response to Reviewer SUPh (1/2)**
>
> > W1: The paper mentions but does not compare against other approaches. FlexMDM and DDOT are both applicable to infilling and can be directly compared with, and their code is available.
>
> Thank you for pointing out these closely related works. We will clarify our relationship to them in the revised version. FlexMDM is a recent concurrent work that proposes a new formulation, also evaluated on infilling tasks, and demonstrates performance gains. However, its performance improvement on HumanEval-Infilling (single-line) remains significantly below ours (FlexMDM achieves pass@1 **~65% with 4096 decoding steps**, whereas **DreamOn reaches > 90% pass@1 within 60 denoising steps** as shown in Figure 4 in the appendix.
>
> DDOT is also a recent work that proposes continuous positional encoding to address variable-length generation, but is evaluated at a smaller scale and its architecture is not compatible with current large diffusion language models such as Dream.  Our work specifically targets improving existing masked diffusion language models without modifying the architectures: DreamOn is a drop-in augmentation that can be applied to any existing masked diffusion language model checkpoint with minimal training, implementing length control only via two additional sentinel states. The training compute needed for DreamOn is only 0.15% compared with the compute for pretraining.
>
> > W2: The paper considers only pass@1. CodeFusion [1—missing reference] shows that code diffusion models excel at higher pass@k rates, because of the model's increased diversity: compiler and execution feedback can be used to select candidates.
>
> Thanks for pointing out CodeFusion and for suggesting evaluating Pass@k to better assess diversity. We will add CodeFusion [1] to the manuscript . Our current experiments focus on Pass@1 to isolate the effect of length control of DreamOn, showing that DreamOn improves significantly upon existing diffusion language models.
>
> DreamOn is fully compatible with multi-sample decoding and Pass@k metrics, where for each prefix-suffix pair, we can sample multiple diffusion trajectories (including different length-adjustment paths) and compute Pass@k rates. The results are shown below:
>
> | Mask Length | 4    | 8    | 16   | 32   | 64   |
> |-------------|------|------|------|------|------|
> | Pass@1 (temp=0.6) | 87.5 | 88.5 | 90.5 | 90.3 | 91.8 |
> | Pass@5 (temp=0.6) | 91.1 | 91.6 | 92.5 | 93.7 | 93.7 |
>
> [1] Singh, M., Cambronero, J., Gulwani, S., Le, V., Negreanu, C. S., & Verbruggen, G. (2023, November). Codefusion: A pre-trained diffusion model for code generation. In The 2023 Conference on Empirical Methods in Natural Language Processing.
>
> > Q1: How does the performance evolve in function of the length of the final generation? I'm supposing that longer masks, which require more [insert], will perform worse?
>
> Thanks for raising this point. Our current results provide evidence that DreamOn substantially reduces the sensitivity seen in the base diffusion language models. There is a mild improvement when using longer mask length which we attribute to having enough “convas” to accommodate most solutions, but the gains over the base diffusion language models are larger across all mask lengths.
>
> To address your question more directly, we analyze the accuracy for DreamCoder+DreamOn for different oracle lengths. While our method mitigates the performance degradation caused by length mismatch, there is still a notable gap compared to the oracle performance in the longest range ([64,∞)), indicating room for improvement.
>
> | Answer Length        | [0,4] | (4,8] | (8,16] | (16,32] | (32,64] | [64,∞) |
> |----------------------|-------|-------|--------|---------|---------|--------|
> | init mask=4          | 97.4  | 93.3  | 76.9   | 57.6    | 37.6    | 11.4   |
> | init mask=8          | 95.9  | 95.1  | 85.5   | 63.4    | 40.5    | 12.5   |
> | init mask=16         | 96.9  | 94.9  | 88.5   | 69.8    | 43.1    | 13.5   |
> | init mask=32         | 97.0  | 95.2  | 88.8   | 76.0    | 50.1    | 16.3   |
> | init mask=64         | 96.4  | 94.5  | 88.5   | 75.5    | 55.3    | 22.5   |
> | oracle               | 97.5  | 96.2  | 89.2   | 75.5    | 55.5    | 35.4   |

---

> > ### Author Response · Authors · 2025-11-21
> > **Response to Reviewer SUPh (2/2)**
> >
> > > Q2: If the above holds, would extending [insert] into [insert1], [insert2], ..., [insertX] make the process converge in fewer iterations? In table 2, the multi-line performance has a big jump to the oracle performance, indicating that requiring many insertions is a bottleneck for longer code snippets.
> >
> > We appreciate this insightful suggestion. In our current design, each [expand] deterministically expands to two [mask] tokens, and further expansions can be applied iteratively. This choice was motivated by simplicity and stability: the model only needs to learn a single expansion token that can be applied over multiple steps rather than choosing among a larger set of expansion factors in a single decision.
> >
> > We agree that larger or parameterized expansion factors would reduce the number of inference steps and potentially handle very long snippets better. However, this would also enlarge the output space and require careful rebalancing of the training objective to keep expansion / deletion well-behaved. We have added a discussion of this trade-off in the updated manuscript and highlight richer expansion vocabularies as a promising direction for future work.

---

### Official Review · Reviewer_bMyK · 2025-10-31

**Soundness:** 2
**Presentation:** 3
**Contribution:** 2
**Rating:** 2
**Confidence:** 3

**Summary:**

This paper identifies the fixed-length mask limitation of diffusion language models for code infilling and proposes DREAMON, a lightweight augmentation that enables dynamic, variable-length generation. DREAMON trains diffusion models to predict two special sentinel states, expand and delete, so the model can autonomously lengthen or shorten masked spans at inference without architectural changes. The authors demonstrate large gains on code-infilling benchmarks, showing near-oracle accuracy and much faster, more robust generation.

**Strengths:**

1. The paper proposes a clear mechanism (expand/delete sentinel states) that gives masked diffusion models native variable-length control without changing model architecture.

2. Empirical results show strong improvements on code-infilling benchmarks, achieving near-oracle accuracy and robust generation compared to prior diffusion and autoregressive baselines.

**Weaknesses:**

1. I feel the baselines are weak — simply plugging this into a DLM will almost certainly yield improvements. You should compare against other methods applied to DLMs to see whether DREAMON still has an advantage. Or are you the first to implement dynamic-length generation on diffusion language models?

2. Claiming in Contribution 1 and 2 that you’ve solved the fixed-length bottleneck of DLMs feels overstated, because the validation is only on code infilling. If you want to make that claim, you need to validate on a more diverse set of tasks.

3. As you note in lines 466–469, your advantages over concurrent work are low training cost and no architectural changes; however, the DAEDAL method you cite in lines 464–465 is training-free. I know that is concurrent work, but the advantage is not accurately stated.

**Questions:**

I didn’t have time to read all the details carefully.
1. In a single inference run, is the length that `[expand]` produces a fixed value? If so, what happens if fewer tokens are actually needed at that point, and how sensitive are results to this parameter?

2. Is it possible for `[expand]` to expand into `[delete]` (i.e., can the expansion produce delete sentinels)?

3. Have the 7B DLMs already been SFT-trained on OpenCoder, and are the training settings the same when you add DREAMON? The single-line improvement looks unusually large — how should that be explained?

---

> ### Author Response · Authors · 2025-11-21
> **Response to Reviwer bMyK (1/2)**
>
> > W1: I feel the baselines are weak — simply plugging this into a DLM will almost certainly yield improvements. You should compare against other methods applied to DLMs to see whether DREAMON still has an advantage. Or are you the first to implement dynamic-length generation on diffusion language models?
>
> We appreciate the reviewer’s comment. While it is intuitive that augmenting DLMs with variable-length capabilities should improve performance, achieving near-oracle performance is not trivial, as shown in Table 2. To our knowledge, there is very limited prior work that systematically studies variable-length generation for DLMs at scale. Several prior or concurrent works tackle length control, but as we discuss in the related work section, they are either limited to small-scale experiments or require complicated architecture changes and more complex pipelines.
>
> > W2: Claiming in Contribution 1 and 2 that you’ve solved the fixed-length bottleneck of DLMs feels overstated, because the validation is only on code infilling. If you want to make that claim, you need to validate on a more diverse set of tasks.
>
> We agree with the reviewer that our current evaluation primarily focuses on code infilling, as reflected in the title. We evaluate code infilling because it is a realistic and highly length-sensitive setting, where fixed-mask diffusion models exhibit sharp degradation when the mask length mismatches the target (Tables 1 & 2). This makes it an ideal testbed for studying the variable-length problem systematically. That said, we acknowledge that the current scope is limited; as noted in the conclusion, we intend to extend DreamOn to more general tasks in future work.
>
> To address your concern, we evaluate our method on ROCStories[1] to show its application in prefix-guided generation. Specifically, we train the model to *generate the last sentence of a story* and report the Rouge-L score for different mask lengths.
> - Dream + SFT: Dream-7B trained on ROCStories without [expand] or [delete] tokens
> - DreamOn: Dream-7B trained on ROCStories with [expand] or [delete]
>
> | Rouge-L  |    4 Mask |    8 Mask |   16 Mask |   32 Mask |
> |--------|-----|-----|-----|-----|
> | Dream + SFT  | 16.3 | 25.1 | 22.4 | 16.7 |
> | DreamOn | 24.5 | 31.4 | 31.3 | 30.6 |
>
> The results in the table show that while the Dream performance significantly fluctuates over the mask length, DreamOn's performance remains consistently high and stable across all mask lengths. This demonstrates that DreamOn successfully decouples generation quality from the initial length, generalizing to variable-length generation for prefix-guided tasks.  We will incorporate this into the revised manuscript.
>
> [1] Mostafazadeh, Nasrin, et al. "A corpus and cloze evaluation for deeper understanding of commonsense stories." Proceedings of the 2016 Conference of the North American Chapter of the Association for Computational Linguistics: Human Language Technologies. 2016.
>
> > W3: As you note in lines 466–469, your advantages over concurrent work are low training cost and no architectural changes; however, the DAEDAL method you cite in lines 464–465 is training-free. I know that is concurrent work, but the advantage is not accurately stated.
>
> Thank you for pointing this out. The concurrent work DAEDAL is indeed training-free, but it relies on a complicated two-stage inference pipeline with multiple confidence-based thresholds; in addition, it does not directly evaluate fine-grained length control in infilling-style settings. In contrast, DreamOn enables native length control with an inference procedure almost identical to vanilla masked DLMs, aside from introducing two additional tokens. Our method also does not rely on hyperparameters such as thresholds in DAEDAL.
>
> > Q1: In a single inference run, is the length that [expand] produces a fixed value? If so, what happens if fewer tokens are actually needed at that point, and how sensitive are results to this parameter?
>
> Yes, each [expand] token always deterministically expands into two [mask] tokens in one denoising step, giving straightforward and stable control over the length. If expansions yield more tokens than needed during generation, the model can subsequently predict [delete] tokens to contract the sequence; if more space is needed, the model can also predict [expand] tokens in the following denoising steps.
>
> > Q2: Is it possible for [expand] to expand into [delete] (i.e., can the expansion produce delete sentinels)?
>
>  An [expand] token always produces two [mask] tokens, so it will never directly produce [delete]. The resulting mask tokens may be predicted as [delete] in later denoising steps, allowing the model to flexibly refine the length as needed.

---

> > ### Author Response · Authors · 2025-11-21
> > **Response to Reviewer bMyK (2/2)**
> >
> > > Q3: Have the 7B DLMs already been SFT-trained on OpenCoder, and are the training settings the same when you add DREAMON? The single-line improvement looks unusually large — how should that be explained?
> >
> > Our base models are not trained on OpenCoder. To investigate the influence of training data, we trained an infilling model on the same data without either expansion or deletion mechanisms. The results for single-line infilling are as follows:
> >
> > | Mask Length | 4    | 8    | 16   | 32   | 64   | Avg  | Oracle |
> > |-------------|------|------|------|------|------|------|--------|
> > | DreamCoder-7B | 24.9 | 61.2 | 72.6 | 62.4 | 55.5 | 55.3 | 93.3   |
> > | DreamOn       | 88.7 | 90.6 | 91.0 | 91.6 | 92.1 | 90.8 | 92.1   |
> > | SFT (w/o Expand w/o Delete) | 25.7 | 57.4 | 61.8 | 51.3 | 49.1 | 49.1 | 93.9   |
> >
> > We find training on oracle-length infilling only improves the oracle-length performance and causes performance decrease for other mask lengths. Therefore, the gap is mostly due to the length sensitivity. It simply indicates that vanilla diffusion lms have the capacity to handle infilling tasks well; they are just limited by the fixed-length issue. DreamOn aims to address this issue by enabling the model to adapt its length during generation.

---

> > > ### Comment · Reviewer_bMyK · 2025-11-28
> > >
> > > Thank you for the author's explanation, which resolved most of my concerns. I will raise the score to 6.

---

### Official Review · Reviewer_Pntm · 2025-11-01

**Soundness:** 2
**Presentation:** 2
**Contribution:** 3
**Rating:** 6
**Confidence:** 4

**Summary:**

The paper proposes DreamOn, a simple extension to diffusion language models that enables variable-length generation by introducing [expand] and [delete] control tokens. It eliminates the need for fixed mask lengths, significantly improving infilling performance on code benchmarks and matching oracle-length results.

**Strengths:**

- Addresses a critical and practical limitation of diffusion language models—fixed-length generation—with a minimal, plug-and-play solution.
- Achieves strong empirical results, closing the performance gap with autoregressive models and matching oracle-length performance on standard infilling benchmarks. Table 1's performance is impressive.
- Requires no architectural changes and integrates seamlessly into existing DLMs with only minor modifications to the training objective.

**Weaknesses:**

- Clarity of Figure 2: The current illustration of the diffusion process in Figure 2 is hard to follow. It would be significantly clearer if the forward and backward processes were presented separately, with explicit depiction of how [expand] and [delete] tokens interact with the sequence during denoising.
- Inconsistent or Missing Baselines: The paper mentions evaluating LLaDA (line 305) but does not include it in any experimental results (e.g., Table 1 or 2), making it difficult to assess comparative performance. Additionally, there appears to be an inconsistency between Table 1 and Table 2: Table 1 claims all experiments use an initial mask length of 64, yet the performance reported for mask length = 64 in Table 2 does not align with the corresponding numbers in Table 1.
- Lack of Analysis on Length Control Stability: While [expand] and [delete] tokens enable dynamic length adjustment, the paper does not discuss mechanisms to prevent unstable behaviors, such as oscillation (e.g., expanding a segment only to delete it in the next step) or unbounded expansion.

**Questions:**

- Could the authors clarify the discrepancy between Table 1 and Table 2 regarding the mask length = 64 setting?
- Is there any safeguard (e.g., termination condition, token-level constraints, or training regularization) to avoid infinite loops or erratic length changes caused by conflicting [expand]/[delete] predictions?
- The method is evaluated only on infilling tasks. Can DreamOn be extended to general unconditional or prefix-guided generation tasks, where target length is also unknown a priori?
- Would it be possible to report the actual FLOPs or computational cost in Table 1 to better assess the efficiency trade-offs of dynamic length adaptation?

---

> ### Author Response · Authors · 2025-11-21
> **Response to Reviewer Pntm (1/2)**
>
> > W1. Clarity of Figure 2: The current illustration of the diffusion process in Figure 2 is hard to follow. It would be significantly clearer if the forward and backward processes were presented separately, with explicit depiction of how [expand] and [delete] tokens interact with the sequence during denoising.
>
> Thanks for your advice. We have revised our figure to improve its clarity. Please kindly refer to the new manuscript.
>
>
> > W2: Inconsistent or Missing Baselines: The paper mentions evaluating LLaDA (line 305) but does not include it in any experimental results (e.g., Table 1 or 2), making it difficult to assess comparative performance. Additionally, there appears to be an inconsistency between Table 1 and Table 2: Table 1 claims all experiments use an initial mask length of 64, yet the performance reported for mask length = 64 in Table 2 does not align with the corresponding numbers in Table 1. & Q1: Could the authors clarify the discrepancy between Table 1 and Table 2 regarding the mask length = 64 setting?
>
> Thank you for pointing out this inconsistency. We did evaluate LLaDA-8B under the same evaluation protocol. We inadvertently omitted this row from Table 1. We have added it in the revised version. The performance of LLaDA is as follows:
>
> | | Single-Line | Multi-Line | Santa-Coder|
> |-|-|-|-|
> | LLaDA|48.3|21.1|35.1|
>
>
> The performance in Table 2 is based on DreamCoder-7B. The results with mask length 64 align with Table 1, 92.1 for single-line infilling, 57.1 for multi-line infilling, and 79.0 for SantaCoder-FIM.
>
> > W3: Lack of Analysis on Length Control Stability: While [expand] and [delete] tokens enable dynamic length adjustment, the paper does not discuss mechanisms to prevent unstable behaviors, such as oscillation (e.g., expanding a segment only to delete it in the next step) or unbounded expansion. & Q2:Is there any safeguard (e.g., termination condition, token-level constraints, or training regularization) to avoid infinite loops or erratic length changes caused by conflicting [expand]/[delete] predictions?
>
> During inference, the primary failure mode we observed was unbounded expansion. To mitigate this, we incorporate two practical safeguards as discussed in L288-L289 in our manuscript:
> 1. Maximum generation length: The intermediate (middle) sequence is capped at a predefined maximum token count. Once this limit is reached, mask expansion is disabled, preventing further growth.
> 2. Maximum expansion steps: The number of expansion operations is limited to a fixed number E. After E expansions, the model can no longer trigger [expand], ensuring termination.
>
> In our experiments, we did not observe infinite loops or erratic length oscillations, so we did not implement explicit cycle detection for such cases. However, we acknowledge that more sophisticated control mechanisms could further enhance robustness, and we plan to implement these features in future work.
> We do not implement a safeguard for infinite loops, as it is not observed in our model. We will explore more delicate control mechanisms in our future work.
>
> > Q3:The method is evaluated only on infilling tasks. Can DreamOn be extended to general unconditional or prefix-guided generation tasks, where target length is also unknown a priori?
>
> We thank the reviewer for this thoughtful question. While our current evaluation focuses on infilling tasks, this choice is deliberate: as illustrated in Figure 1 of our paper, code infilling is particularly sensitive to mismatches between the target sequence length and the initially provided mask length, making it an ideal benchmark for evaluating variable-length generation capabilities.
>
> We evaluate our method on ROCStories[1] to show its application in prefix-guided generation. Specifically, we train the model to *generate the last sentence of a story* and report the Rouge-L score for different mask lengths.
> - Dream + SFT: Dream-7B trained on ROCStories without [expand] or [delete] tokens
> - DreamOn: Dream-7B trained on ROCStories with [expand] or [delete]
>
> | Rouge-L  |    4 Mask |    8 Mask |   16 Mask |   32 Mask |
> |--------|-----|-----|-----|-----|
> | Dream + SFT  | 16.3 | 25.1 | 22.4 | 16.7 |
> | DreamOn | 24.5 | 31.4 | 31.3 | 30.6 |
>
> The results in the table show that while the Dream performance significantly fluctuates over the mask length, DreamOn's performance remains consistently high and stable across all mask lengths. This demonstrates that DreamOn successfully decouples generation quality from the initial length, generalizing to variable-length generation for prefix-guided tasks.  We will incorporate this into the revised manuscript.
>
> [1] Mostafazadeh, Nasrin, et al. "A corpus and cloze evaluation for deeper understanding of commonsense stories." Proceedings of the 2016 Conference of the North American Chapter of the Association for Computational Linguistics: Human Language Technologies. 2016.

---

> > ### Author Response · Authors · 2025-11-21
> > **Response to Reviewer Pntm (2/2)**
> >
> > > Q4: Would it be possible to report the actual FLOPs or computational cost in Table 1 to better assess the efficiency trade-offs of dynamic length adaptation?
> >
> > Thanks for your question. Dream-7B is pretrained on 580B tokens with 24576 GPU hours on H800 while DreamOn only requires 40 GPU hours to enable existing diffusion models to perform variable-length generation. **The compute of DreamOn is only 0.15% compared with the compute for pretraining.**
> >
> > We have clarifies this in our updated manuscript to emphasize the compute of our adaptation method compared with pretraining.

---

### Official Review · Reviewer_4aWj · 2025-11-01

**Soundness:** 3
**Presentation:** 3
**Contribution:** 3
**Rating:** 6
**Confidence:** 4

**Summary:**

Diffusion language models have been a promising direction, especially in code generation because of flexible code in-filling capabilities. However, traditional diffusion language models use a fixed size canvas and diffuse language tokens within those. This paper introduces a novel way to remove the constraints of a fixed size canvas with two special control tokens, [expand] and [delete], allowing the model to autonomously adjust the output length during generation without architectural changes. Experiments show this method removes the fixed-length bottleneck, allowing DLMs to achieve performance on par with state-of-the-art autoregressive models and nearly matching oracle performance where the correct length is known in advance.

**Strengths:**

1. This paper  addresses the fixed-length mask, a major practical limitation for diffusion language models with `[expand]` and `[delete]` token solution is elegant, effective, and requires no architectural changes.

1. The paper has very nice presentation, with illustrative examples of the control tokens, and their effects.

1. The paper clearly validates the necessity of both the expansion and deletion mechanisms through detailed experiments and ablations.

1. The proposed method allows diffusion language models to match or exceed the performance of state-of-the-art autoregressive models in code infilling of the same compute level.

**Weaknesses:**

- The method is presented as a general solution for Diffusion Language Models, but it is only evaluated on Python code infilling. It is unclear if the `[expand]` and `[delete]` logic would translate well to more fluid and creative natural language tasks.

- The deletion broadcasting is an example of a "training-free" efficiency, but it seems like it would be better suited if this were learned in a data-driven manner.

**Questions:**

- For deletion broadcasting, were there any experiments done to make the model learn the start and ends of spans for deletions instead of the heuristic of deleting all mask tokens?

- I'm curious how your method would behave on non-structured language modeling tasks, would it be fair to say that using the expansions and deletions would be a more significant challenge when it's not code?

---

> ### Author Response · Authors · 2025-11-21
> **Response to Reviewer 4aWj**
>
> > W1: The method is presented as a general solution for Diffusion Language Models, but it is only evaluated on Python code infilling. It is unclear if the [expand] and [delete] logic would translate well to more fluid and creative natural language tasks. & Q2: I'm curious how your method would behave on non-structured language modeling tasks, would it be fair to say that using the expansions and deletions would be a more significant challenge when it's not code?
>
> We initially focused our evaluation on Python code infilling because its performance is exceptionally sensitive to the mismatch between the initial mask length and the oracle length, as shown in Figure 1. And infilling segments can be easily verified with a code interpreter. This characteristic makes it an excellent, realistic, and demanding testbed for our method.
>
> To directly address the behavior on more fluid and creative language tasks, we evaluated DreamOn on the ROCStories corpus [1], which requires a high degree of commonsense and narrative understanding.
> We evaluate our models by *infilling one sentence in the story*. We report Rouge-L scores in the table below. We have added this experiment to our manuscript.
>
> - Dream + SFT: Dream-7B trained on ROCStories without [expand] or [delete] tokens
> - DreamOn: Dream-7B trained on ROCStories with [expand] or [delete]
>
> | Rouge-L  |    4 Mask |    8 Mask |   16 Mask |   32 Mask |
> |--------|-----|-----|-----|-----|
> | Dream + SFT  | 19.2 | 29.8 | 26.5 | 18.9 |
> | DreamOn | 31.6 | 31.4 | 31.3 | 30.6 |
>
> DreamOn successfully generalizes to this non-structured domain, achieving superior and highly consistent Rouge-L scores regardless of the initial mask length. This performance significantly surpasses the Dream baseline across all lengths, particularly when the initial mask is extremely short (length 4) or long (length 32).
>
> Our experiment confirms that the length-adaptive logic of DreamOn is fully effective in managing the variable lengths inherent in creative natural language generation, demonstrating its strong generalizability beyond code.
>
> [1] Mostafazadeh, Nasrin, et al. "A corpus and cloze evaluation for deeper understanding of commonsense stories." Proceedings of the 2016 Conference of the North American Chapter of the Association for Computational Linguistics: Human Language Technologies. 2016.
>
>
> > W2: The deletion broadcasting is an example of a "training-free" efficiency, but it seems like it would be better suited if this were learned in a data-driven manner. & Q1: For deletion broadcasting, were there any experiments done to make the model learn the start and ends of spans for deletions instead of the heuristic of deleting all mask tokens?
>
> Thanks for your question. We agree that length prediction in deletion broadcasting could be learned via a data-driven approach, such as by introducing a regression head to predict output length.
>
> However, our primary goal is to enable variable-length generation with the minimal possible alteration of existing diffusion models. We specifically design deletion broadcasting as a training-free method that requires no architectural modifications. This maintains simplicity, ensures immediate compatibility with any existing diffusion model, and avoids the need for adding new modules. The compute of DreamOn to enable variable-length generation is only 0.15% compared with the compute for pretraining.
>
> Furthermore, our method implicitly learns the optimal deletion behavior efficiently. During training, all [delete] tokens following the middle sequence are masked. This setup naturally encourages the model to implicitly learn the optimal termination point by predicting the [delete] token as the signal for the end of the infilling span. Therefore, instead of relying on a heuristic to delete all mask tokens, the model's prediction of a [delete] token acts as a learned, data-driven termination signal for the diffusion step. This preserves the elegance and generality of our approach while achieving the desired adaptive length control.

---

### Author Response · Authors · 2025-11-26
**General Response**

We thank the reviewers for their valuable feedback. We have made a thorough revision to our manuscript.

**Thank you for acknowledging our strengths:**

1. An elegant, plug-and-play mechanism using [expand] and [delete] tokens that enables variable-length generation in diffusion language models without any architectural changes (Reviewers 4aWj, Pntm, bMyK).

2. DreamOn achieves strong empirical performance, matching autoregressive models and reaching near-oracle accuracy in code infilling (Reviewers 4aWj, Pntm, bMyK).

3. Thorough ablations and illustrative examples that validate design choices (Reviewers 4aWj, SUPh).

**Our Key Revisions**：

1. We add experiment results on the ROCStories dataset that DreamOn can achieve stable, high-quality infilling and prefix-guided generation in story writing across diverse initial mask lengths. It demonstrates that **DreamOn generalizes beyond code infilling to fluid natural language infilling and generation**.

2. We revised Figure 2 to clearly separate forward and backward processes of DreamOn for better readability.

3. We emphasize the advantages of DreamOn compared with concurrent work.
    - **Minimal training compute**: DreamOn only requires 0.15% of the compute needed for pretraining Dream-7B.
    - **Superior performance and flexible infilling**: DreamOn achieves > 90% pass@1 for single-line infilling within 60 steps, compared to the concurrent work FlexMDM’s 65% pass@1 with 4096 steps.

We thank the reviewers again for their time and constructive comments in reviewing our work. We are happy to address any further questions.

---

### Meta-Review · Area_Chair_i4Z8 · 2026-01-07

**Summary:**

DreamOn introduces a simple yet effective mechanism to enable variable-length generation in Diffusion Language Models (DLMs) through two control tokens ([expand] and [delete]), removing a fundamental limitation without architectural changes.

Reviewers recognized the elegance and strong empirical results but raised concerns about evaluation scope (limited to code) and baseline comparisons with concurrent methods, etc. The rebuttal substantially addressed these concerns by adding ROCStories experiments demonstrating natural language generalization and clarifying computational efficiency (0.15% of pretraining cost). One reviewer (bMyK) raised the score from 2 to 6 after the rebuttal, so the scores after rebuttal are 6, 6, 6, 4.

**Reviewer Concerns:**

Concerns fully addressed:
- Natural language generalization (4aWj W1, bMyK W2): Authors added ROCStories experiments showing consistent Rouge-L performance (30.6-31.6) across all mask lengths, outperforming baselines and demonstrating robust generalization beyond code.
- Baseline comparisons with concurrent methods (bMyK, SUPh): Authors explained advantages over FlexMDM (>90% pass@1 in 60 steps vs ~65% in 4096 steps) and DDOT (different scale, incompatible architecture), but did not provide direct experimental comparison. Response is reasonable given concurrent publication and architectural incompatibility.
- Figure clarity (Pntm): Figure 2 revised to separate forward/backward processes.
- Stability mechanisms (Pntm): Authors confirmed safeguards exist (max generation length, max expansion steps E) and reported no observed oscillation issues.

Concerns partly addressed:
- Pass@k evaluation (SUPh): Authors provided Pass@5 results (91.1-93.7% vs 87.5-91.8% Pass@1), showing improved diversity. It addressed the concern but could be more comprehensive.
- Performance degradation on very long sequences (SUPh): Authors' analysis shows notable performance gap for oracle length $[64,\infty)$ even with DreamOn (22.5% vs 35.4% oracle), indicating limitations for extremely long infilling remain unresolved.

**Reviewer Scores:**

- Reviewer 4aWj: Likely remains 6.
- Reviewer Pntm: Likely 6 or 8.
- Reviewer bMyK: Confirmed updating to 6 (from 4).
- Reviewer SUPh: Likely 4 or 6.

---

### Decision · Program_Chairs · 2026-01-26

Accept (Poster)